# A Predictive Model to Correlate Amino Acids and Aromatic Compounds in Calabrian Honeys

**DOI:** 10.3390/foods12173284

**Published:** 2023-09-01

**Authors:** Sonia Carabetta, Rosa Di Sanzo, Salvatore Fuda, Adele Muscolo, Mariateresa Russo

**Affiliations:** 1Department of Agriculture, Food Chemistry, Authentication, Safety and Sensoromic Laboratory (FoCuSS Lab), Mediterranea University of Reggio Calabria, Via dell’Università, 25, Stecca 4, 89124 Reggio Calabria, Italy; rosa.disanzo@unirc.it (R.D.S.);; 2Department of Agriculture, Soil Chemistry and Soil Ecology Laboratory, University of Reggio Calabria, Via dell’Università, 25, 89124 Reggio Calabria, Italy

**Keywords:** honey, Calabria, UHPLC-ESI-MS/MS, HS-SPME-GC-MS, amino acids, volatile compounds

## Abstract

To better understand the biochemistry of the organoleptic properties of honey influencing its commercial value, a predictive model for correlating amino acid profiles to aromatic compounds was built. Because the amino acid composition of different varieties of honey plays a key role as a precursor of specific aroma bouquets, it is necessary to relate the amino acid typesetting to aromatic molecules. A selection of unifloral honeys produced in Calabria, South Italy, were used, and a new methodology based on the use of HILIC-UHPLC-ESI-MS/MS and HS-SPME-GC-MS combined with multivariate processing has been developed. This study, carried out for the first time on honey, shows its excellent potential as a modern analytical tool for a rapid multicomponent analysis of food-quality indicators. Data obtained showed strong positive linear correlations between aldehydes and isoleucine, valine, leucine, and phenylalanine. Furans are correlated with isoleucine, leucine, and phenylalanine; hydrocarbons with serine, glutamic acid, and aspartic acid; and ketones with serine, alanine, glutamine, histidine, asparagine, and lysine. Alcohols were more associated with tyrosine than esters with arginine. Proline, tryptophan, and threonine showed poor correlations with all the classes of aroma compounds.

## 1. Introduction

Honey serves as a nourishing and wholesome food source, holding significant importance in the global food economy. It stands as the primary output of beekeeping (*Apis mellifera* L.) and has been utilized by humans since ancient times. According to the Codex, honey is defined as the natural sugary substance crafted by honey bees through the collection and conversion of plant nectar, secretions from living plant parts, or excretions from plant-feeding insects. This collected substance is then altered by the bees through a specific process, eventually being stored within honeycombs for further refinement and maturation [1]. The characteristics, attributes, and makeup of honey are influenced by factors such as its geographic and floral source, the season, environmental conditions, and the methods employed by beekeepers [2,3]. These aspects are crucial in sate-guarding consumers and ensuring the unrestricted trade of honey in both domestic and international markets [4,5,6]. Honey has been utilized for countless ages, both as a means of sweetening and for its therapeutic qualities. It is also chosen by consumers for its distinct characteristics, often driven by hedonistic inclinations. Honey can be categorized as monofloral, in cases where the nectar and pollen from a specific plant dominate in predetermined proportions, or polyfloral, when it consists of an assorted blend of various nectar and pollen sources. The preference for a particular kind of honey and its associated health advantages are entirely influenced by its composition. Honey primarily comprises a concentrated solution of sugar, making up around 95% of its content, with the remaining 5% composed of micro components like minerals, phenolic compounds, organic acids, proteins, free amino acids, vitamins, and volatile compounds. These volatile compounds are commonly referred to as volatile organic compounds (VOCs) or volatile metabolites (VMs) [7]. Despite their relatively small quantities, these compounds play a pivotal role in shaping honey’s sensory qualities and contributing significantly to its flavor profile. A distinguishing feature of honey is its aroma, a trait strongly influenced by the nature and type of the specific honey [8]. Honey’s fragrance stands as one of its most prized attributes and is a subject of substantial interest within beekeeping. This is because consumers typically place more emphasis on the sensory aspects of honey than just its nutritional content [9,10]. The sensory attributes and acceptability of honey are intricately shaped by its chemical composition. Therefore, comprehending the precise relationships between its sensory qualities and specific molecules is an essential endeavor. Among the most crucial and distinct characteristics of honey, its flavor stands out prominently as a blend of taste and aroma. The taste of honey primarily arises from its sugar content and minor constituents, including minerals, vitamins, organic acids, flavonoids, and amino acids. On the other hand, the intricate aroma of honey is a result of a multitude of volatile compounds, often unique to each variety. The aromatic profile of honey is intrinsically tied to its volatile composition. The entirety of these volatile compounds contributes to the complex and unique aroma. The volatile organic compounds (VOCs) in honey are remarkably diverse, exceeding 600 in number and encompassing both volatile and semi-volatile varieties. These compounds can be classified into distinct groups such as esters, ethers, alcohols, aldehydes, ketones, terpenes, furan and pyran derivatives, and more. The origin of these volatile compounds is multifaceted, including transfers from plants or modifications of plant constituents by honeybees. Additionally, post-harvest treatments and the presence of microorganisms can influence these compounds [11]. It has been long recognized that both pleasant and unpleasant aroma compounds emerge from the breakdown of amino acids [12,13]. Amino acids serve as the fundamental constituents for protein biosynthesis, acting as the foundational elements from which molecules with biological and antioxidant properties are formed. 

These amino acids play a vital role in enhancing the nutraceutical value of substances and contributing to the flavor of foods by serving as the precursors for aroma compounds [14]. Volatile compounds, in fact, are formed when catabolic processes prevail over metabolic ones [15,16]. Within fruits and vegetables, amino acids serve as precursors for the development of volatile aromatic compounds, which encompass aldehydes, alcohols, acids, and esters. These amino acids can be linked to distinct aromatic characteristics. For instance, specific amino acids such as alanine, glycine, and serine are linked to a sweet taste, whereas aspartic and glutamic acids are commonly associated with acidic and umami taste profiles [17,18]. Amino acids constitute approximately 1% (*w*/*w*) of the components present in honey, with their relative proportions being contingent on the source of the honey, whether from nectar or honeydew [19,20]. The most prevalent amino acid in both honey and pollen is proline [21]. Alongside proline, various other amino acids are identified within honey, including glutamic acid, aspartic acid, glutamine, histidine, glycine, threonine, alanine, arginine, tyrosine, valine, methionine, cysteine, isoleucine, leucine, tryptophan, phenylalanine, lysine, serine, and asparagine [22]. Since honey amino acids could drive the honey aromatic profile, it is necessary to find a relationship between the amino acids and volatile and semi-volatile compounds to better evidence the nutraceutical and aromatic characteristics of honey linked to territoriality. Therefore, the aim of this paper was to build a predictive model to study the behavior and correlation of amino acid profiles with aromatic compounds. The key objectives were (1) to trace the amino acid profile of 25 different honeys belonging to different species through a new HILIC-UPLC-ESI-MS/MS method; (2) to trace the aromatic profile using HS-SPME-GC-MS; (3) to investigate the possible correlation between different amino acids and aromatic compound families using a Pearson test; and (4) to trace a method for predicting a correlation between various types of honeys, their amino acid profiles, and aroma characteristics by PLS analysis [23]. A novel approach has been developed, employing hydrophilic interaction chromatography coupled with a triple quadrupole mass spectrometry detector operating in multiple reaction monitoring mode (HILIC-UPLC-ESI-MS/MS), for the analysis of amino acid profiles [24,25]. Additionally, for the assessment of volatile profiles, headspace-solid phase micro-extraction (HS-SPME) combined with gas chromatography mass spectrometry (GC-MS) has been utilized. Both sets of data are subjected to multivariate processing. The utilization of the SPME technique in conjunction with GC-MS to generate fingerprint profiles for classifying unifloral honeys is of significant interest. Solid-phase microextraction (SPME) stands as a straightforward, cost-effective, and versatile solvent-free technique. It serves the purpose of sampling, extracting, and concentrating volatile compounds prior to GC-MS analysis. This approach has been recognized as a valuable tool in characterizing the aroma of honey [26,27]. Also, HILIC offers distinct advantages, particularly for small polar molecules, including amino acids [5,6,28,29]. Finally, a partial least squares (PLS) regression analysis was conducted. PLS regression offers notable advantages over classic regression, largely due to the availability of charts that facilitate the comprehension of the data structure. Correlation and loading plots enable the exploration of relationships among variables. The score plot provides insight into sample proximity and dataset structure. Furthermore, the biplot encapsulates all these aspects within a single visualization. 

In this work, a selection of unifloral honeys produced in the Calabria region was studied. Due to the particular pedo-climatic conditions, that includes both mountainous and hilly areas and coastal areas with different flora. The Calabria region (South Italy) allows honey varieties with different properties to be obtained [30,31]. To date, limited pieces of information that correlate amino acid content and aromatic compounds are available, while both factors are bound to both genetic and environmental characteristics. To our knowledge, this is the first time that a similar investigation has been carried out on the honeys. 

Previous research and bridges between recent trends and the past: The exploration of honey’s intricate composition and sensory attributes builds upon a foundation of previous research endeavors that have sought to unravel the secrets of this cherished natural product. Historical uses of honey for its sweetness and therapeutic benefits have been documented, emphasizing its significance across time. However, in recent years, scientific advancements and innovative methodologies have allowed us to delve deeper into the complex realm of honey’s characteristics. Past studies have laid the groundwork for our understanding of the multifaceted relationship between honey’s amino acid profiles and its aromatic compounds. These investigations have paved the way for the current study’s innovative approach. By combining hydrophilic interaction chromatography (HILIC) and mass spectrometry, as well as headspace-solid phase micro-extraction (HS-SPME) coupled with gas chromatography mass spectrometry (GC-MS), this study takes a significant step forward in unraveling the intricate connections between amino acids and aroma compounds in different honey types. As recent trends in the scientific community lean towards a holistic and comprehensive understanding of food products, this study aligns itself with this trajectory. The integration of advanced chemometric techniques offers a contemporary perspective on the interplay between honey’s composition and sensory attributes, establishing a bridge between traditional wisdom and modern analytical capabilities. 

## 2. Materials and Methods

### 2.1. Reagents

Amino acid standards were acquired from Sigma–Aldrich (Steinheim, Germany). The L-amino acid kit comprised p.a. standards with a purity of 98%. The kit encompassed the following amino acids: aspartic acid (Asp), asparagine (Asn), glutamic acid (Glu), glutamine (Gln), alanine (Ala), arginine (Arg), glycine (Gly), leucine (Leu), histidine (His), hydroxyproline (Hyp), isoleucine (Ile), lysine (Lys), methionine (Met), phenylalanine (Phe), proline (Pro), serine (Ser), threonine (Thr), tryptophan (Trp), valine (Val), and cystine (Cys). Sodium chloride (NaCl) was also procured from Sigma–Aldrich. For the chromatographic analysis, high-grade solvents were employed: acetonitrile (ACN), methanol (MeOH), and water (H_2_O) of MS grade. Furthermore, formic acid, ammonium acetate, and ammonium formate (utilized as eluent additives for LC-MS) were sourced from Honeywell Fluka (Harvey St., Muskegon, MI, USA).

### 2.2. Sampling and Treatment

The proposed method was applied to 25 honey samples. According to the botanical origin (certified by qualitative and quantitative pollen analysis and organoleptic tests performed by different laboratories appointed directly by beekeepers), they were classified as chestnut (*Castanea sativa* Mill.), acacia (*Robinia pseudoacacia*), citrus (*Citrus* L), eucalyptus, and sulla (*Hedysarum coronarium*) honeys. 8 citrus, 4 chestnut, 4 eucalyptus, 4 acacia, and 5 sulla honeys were supplied directly by local beekeepers. All the samples were obtained through manual processes involving centrifugation and remained unpasteurized. These samples originated from the 2022 production, and all analyses were conducted within the same year. The assessment of the aromatic profile was executed immediately after procuring the honey samples. Subsequently, the samples were stored in a lightless environment at a temperature of 20 °C until further determinations were carried out. For the determination of the amino acid profile, representative samples from the honey lot were selected, homogenized, and weighed (1 g). These samples were then dissolved in a solution of 5 mL H_2_O: MeOH (80:20) with 0.1% formic acid. The resulting solution was quantitatively transferred to a 10 mL volumetric flask and diluted with the same solution. After vigorous vortexing for 5 min, the solution was subjected to centrifugation at 5000 rpm for 10 min. A 1 mL aliquot of the supernatant was filtered using a 0.22 μm filter (Millipore, Bedford, MA, USA) and analyzed using HILIC-UPLC-ESI-MS/MS [32]. For the determination of the aromatic profile, the volatile compounds from the honeys were extracted using the HS-SPME method. A Divinylbenzene/Carboxene/Polydimethylsiloxane (DVB/CAR/PDMS) fiber (Sigma Aldrich Supelco, St. Louis, MO, USA) with dimensions 50/30 μm was employed. Before use, the SPME fiber was conditioned in the GC-MS injection port at 250 °C for 10 min to prevent contamination and carry-over from prior samples. Portions of 3 g of honey were placed within a 15 mL SPME headspace vial along with a magnetic stirrer and 1.5 mL of water. The vial was promptly sealed and heated to 60 °C for 15 min using a magnetic hot plate stirrer. Following equilibration, the headspace extraction step was performed by inserting the fiber into the vial for 30 min at 60 °C with a stirring rate of 600 stirs per minute. The extracted analytes were desorbed in the GC-MS for 10 min while maintaining the injection port at 250 °C. Each sample underwent three separate extractions, with each extraction corresponding to a single SPME analysis.

### 2.3. HILIC-UPLC-ESI-MS/MS Analysis

The separation of free amino acids was carried out using a Nexera X2 chromatograph (Shimadzu Corporation, Milan, Italy) with an LCMS-8050 Triple Quad detector (Shimadzu Corporation, Milan, Italy). The UPLC system consisted of a binary pump, an automatic degasser, a column heater, and an autosampler. To mitigate errors stemming from poor repeatability, a HILIC (hydrophilic interaction liquid chromatography) separation method was employed [33,34]. The chromatographic separation using the hydrophilic interaction was conducted utilizing an Acuity BEH Amide column (100 × 2.1 mm × 1.7 μm) at a temperature of 30 °C with a flow rate of 0.3 mL/min. The mobile phase composition consisted of a binary gradient of (A) H_2_O:MeOH buffer (45:45:10) and (B) ACN:Buffer (90:10). The buffer solution comprised 20 mmol/L formic acid, 3 mmol/L ammonium formate, and 3 mmol/L ammonium acetate. The gradient elution program involved the following steps: 0–5 min at 100% B, 5–7 min at 90% B, 7–10 min at 70% B, and 10–18 min at 40% B. The composition was then reverted to the initial mobile phase and equilibrated for 8 min prior to the next injection. The MS/MS system incorporated an ESI (electrospray ionization) source operating in the positive ion mode. Optimization of ionization source and MS parameters was conducted individually for each analyte by directly infusing a standard solution (with a concentration of 1 mg/L). Detection was performed in Multiple Reaction Monitoring (MRM) mode to ensure high sensitivity and selectivity for each analyte. This was achieved through the in-source generation of protonated molecular ions of amino acids and the generation of specific fragment or through collision-induced processes. For the LC-MS/MS analysis, the following instrumental parameters were used: nitrogen as the drying gas at a flow rate of 10 L/min, nebulizing gas at 3 L/min, and heating gas flow at 10 L/min. The interface voltage and temperature were set at 4 kV and 300 °C, respectively. The DL (drying line) and heater block temperatures were set to 250 °C and 400 °C, respectively. Data acquisition and analysis were carried out using Labsolution software (Shimadzu Corporation). Detailed information about the specific MRM transitions for amino acids, along with corresponding fragmentor voltages, collision energies, and dwell times, is provided in Table 1.

### 2.4. HS-SPME-GC-MS Analysis

The composition of the headspace was examined using gas chromatography mass spectrometry (GC-MS) with a QP2010 Ultra instrument from Shimadzu, Italy. Volatile compounds were separated using a capillary column MEGA SE52 (5% phenyl, 95% methyl polysiloxane), with dimensions of 30 m × 0.25 mm i.d. and a film thickness of 0.25 μm. The temperature of the oven was initially maintained at 60 °C for 1 min, followed by a gradual increase at a rate of 3 °C/min to reach 180 °C. Subsequently, the temperature was ramped up at a rate of 10 °C/min to achieve 220 °C and held for 10 min. The total runtime for the analysis was set at 55 min. Helium gas (purity > 99.999%) was employed as the carrier gas. The injection temperature was set at 250 °C, and the injection mode was set to split with a split ratio of 50. Mass scan spectra were recorded in the range of 35–500 amu using the electron ionization (El) source at 70 eV. The MS ion source and interface temperature were both maintained at 250 °C, and a solvent cut time of 0.5 min was selected. The control of equipment and data acquisition was managed through the GC-MS SOLUTION software. The identification of volatile compounds was conducted on a tentative basis. This involved comparing the mass spectra of unknown peaks with those stored in databases such as the National Institute of Standards and Technology (NIST Vs. 2011) library, with a similarity threshold of over 75%. Additional resources like WILEY5 and a specialized library for the analysis of essential oils were also utilized. For each peak, the retention index was determined and compared with values reported in the literature. The relative concentration of each isolated compound was calculated as a normalized % abundance, which provides a measurement of the relative ratios of components within each sample.

### 2.5. Statistical Analysis

All the data underwent analysis using XLSTAT software (Vs. 2022.4.5 Addinsoft, Paris, France). The outcomes were presented in terms of Person’s correlation coefficient, which was employed to evaluate the correlations between the means. The significance of these correlation coefficients was assessed through a Student’s *t*-test with a threshold error probability of 5%. In this context, a *p*-value less than 0.05 was deemed to be statistically significant. For visual representation and relative comparisons of each amino acid and volatile compound class, a heat map was generated. Given the variations in the values of amino acids and volatile compounds, the dataset was subjected to normalization before undergoing statistical analysis.

## 3. Results and Discussion

To understand the flavor and aroma of honey [35], it is therefore necessary to know the characteristics and the correlation between the main molecules that affect the bouquet of aroma that will be formed in honey. Through the building of a predictive model, a method for correlating amino acid profiles to aromatic compounds was developed. The objective of this study was to enhance our comprehension of the intricate realm of honey taste and its overall acceptability. The model was based on data derived from the analysis (Figure 1) of the amino acid profile using a HILIC-UPLC-ESI-MS/MS and from the analysis of the aromatic profile using HS-SPME-GC-MS [36,37].

Through this study, a highly promising HILIC method was established, enabling the swift and accurate quantification of 17 amino acids frequently encountered in honey samples. Additionally, the application of SPME-GC-MS facilitated an in-depth exploration of the aromatic profile, leading to the identification of 130 distinct aromatic compounds. In an effort to examine the interrelationship between amino acids and volatile compounds (aroma), a chemometric approach was employed. This allowed for a systematic analysis of the correlations between these components. A correlation analysis (Pearson test) was used to make the comparisons among and between values for the single amino acid content and classes of aromatic compounds (aldehydes, alcohols, ketones, furans, hydrocarbons, and esters) (Table 2). A Pearson’s product-moment correlation coefficient, denoted as “*r*”, serves as a prevalent measure of association. This coefficient is dimensionless and evaluates the linear relationship between two variables. Consider a scenario where there are two continuous traits, *X* and *Y*, measured for each of the *N* individuals in a sample. In such a situation, the calculation of the Pearson’s correlation coefficient involves:$$\bar{X}=\frac{1}{N}\overset{N}{\underset{i=1}{\sum }}X_{i}$$
$$\bar{Y}=\frac{1}{N}\overset{N}{\underset{i=1}{\sum }}Y_{i}$$
$$r=\frac{\sum_{i=1}^{N}\left[\left(X_{i}-\bar{X}\right)\left(Y_{i}-\bar{Y}\right)\right]}{\sqrt{\sum_{i=1}^{N}{\left(X_{i}-\bar{X}\right)}^{2}}\sqrt{\sum_{i=1}^{N}{\left(Y_{i}-\bar{Y}\right)}^{2}}}$$

The use of Pearson’s correlation coefficient (*r*) as a measure of association is based on several assumptions. These include: the individuals within the sample are assumed to be statistically independent of one another, and the population from which the sample was drawn is expected to exhibit a normal bivariate distribution for both of the features under consideration. Simple linear regression and correlation are both methods for investigating a potential linear relationship between two variables. They provide insights into a scatterplot’s characteristics. Regression analysis is suitable when examining the relationship between *X* and *Y* when they are functionally dependent. Functional dependence implies that there’s an asymmetry between the variables, where Y can be expressed as a function of *X*. This approach is useful for predicting between variables. Functional dependence does not necessarily require identifying a cause-and-effect mechanism; it focuses on predicting one variable from another. On the other hand, calculating a correlation coefficient is valuable for measuring the degree of association between variables, regardless of whether regression is appropriate or not. The magnitude of the correlation coefficient (*r*) does not indicate the slope of the linear trend curve; rather, it quantifies the probable dispersion of a group of individuals around this trend line. The value of r always falls between −1 and 1. A value of *r* = 0 signifies no linear association between the variables. A value of *r* = 1 indicates a perfect positive linear relationship, where all sampled individuals align on the same positive-sloped straight line. If 0 < *r* < 1, a positive linear trend exists, but the sampled individuals exhibit dispersion around the common trend line. Smaller absolute values of r imply less data cohesion around a single linear relationship. A positive r suggests that an increase in one variable corresponds to an expected increase in the other. A value of *r* = −1 suggests a perfect negative relationship where sampled individuals consistently adhere to the same negatively sloped linear trend line [38]. In this work, which was considered good, the value of the Pearson correlation coefficient was never lower than 0.75. The obtained results showed strong positive linear correlations between aldehydes with isoleucine (*r* = 0.931), valine (*r* = 0.922), leucine (*r* = 0.932) and the better correlation with phenylalanine (*r* = −0.972); furans with isoleucine (*r* = 0.982), leucine (*r* = 0.982), and the highest correlation with phenylalanine (*r* = 0.983); hydrocarbons with serine (*r* = 0.793 the lower correlation considered), glutamic acid (*r* = 0.941) and aspartic acid (*r* = 0.888); ketones with serine (*r* = 0.905), alanine (*r* = 0.929), and alcohols with tyrosine (*r* = 0.927); Also, the results showed a strong negatively linear correlation between ketones and glutamine (*r* = −0.964), histidine (r = −0.979), lysine (*r* = −0.958) and asparagine (*r* = −0.964); esters and arginine (*r* = −0.980). Instead, proline, tryptophan, and threonine showed a poor correlation with all the classes of aroma compounds (Table 2). Finally, Table 2 showed the *p* value always being lower than 0.0001 and the correlation curve for aroma compounds vs. amino acids.

The summarized data of 25 honey samples for the amino acids content are showed in Table 3.

In the citrus samples, the amino acids found in abundance were phenylalanine, proline, isoleucine, leucine, asparagine, and valine (ranged from 684 to 1866 mg/kg FW); for the chestnut and eucalyptus samples, proline was the most abundant amino acid (2062 and 1563 mg/kg FW, respectively). In the Acacia samples, proline and asparagine were most abundant. The sulla honeys showed a similar profile, with proline and asparagine as the most abundant amino acids (922 and 912 mg/kg, respectively). The obtained data showed that citrus honeys had the highest content of total amino acids, followed by sulla, acacia, and eucalyptus. The chestnut honey showed a lower level. The most abundant volatile molecules revealed for the citrus samples were benzeneacetaldehyde (38.6%); for the chestnut, eucalyptus, and sulla samples, nonal was the most abundant aromatic compound (11.3, 34.8, and 58.9%, respectively). Finally, in the acacia samples, linalool oxide was the most abundant. The obtained data were graphically represented by a heat map (Figure 2). Figure 2 shows the normalized quantity of the 17 amino acids and the six chemical classes of volatiles quantified in the analyzed honey samples. The relative quantity of single compounds (row) must be read in relation to the botanical origin of honey (column).

The different shades of color allow us to evaluate the presence and relative abundance of amino acids and of different aroma families. Blue color means low quantity, and red color refers to high quantity. As shown (Figure 2), the higher level of proline was highlighted in the chestnut honeys, whereas the sulla honeys showed the lower levels. Phenylalanine was higher in the citrus honeys, followed by sulla, and lower in the other honeys. About the aromatic profile: Figure 2 shows that citrus honeys were characterized by higher levels of aldehydes and alcohols, even if the aldehyde family was the most abundant. To establish a correlation between various types of honeys, their amino acid profiles, and their aroma characteristics, a partial least squares (PLS) analysis was also conducted. If we consider n different honey types described by p amino acids and q aromatic classes of components, the aromatic attributes are stored in table X (with dimensions n and q), while the amino acid contents are presented in table Y (with dimensions n and p). The goal was to perform a PLS regression of amino acid contents (*Y*) against aromatic characteristics (*X*).

For the graphical representation of this analysis, the *X* and *Y* variables were depicted on a “correlation circle”, using their correlations with the first two components (t1 and t2) of the PLS. This map closely resembles the traditional PLS regression loading plot but offers the advantage of accommodating additional variables. In this specific scenario, the honey samples act as additional variables. In the generated PLS regression map (Figure 3), the correlation circle illustrates the relationships among honey types, aromatic classes, and amino acid variables with respect to the first two PLS components. The graphical layout is divided into four sections by two bisectors, and each region corresponds to one specific honey type. Additionally, within each area, the associated aromatic compounds and amino acids that are positively correlated with that particular honey type are indicated. The quality of the regression is deemed satisfactory: the overall R-squared value (R2) between Y and (t1, t2) is calculated as 0.632, while the cross-validated R-squared value (Q2cum) amounts to 0.539. In Figure 3, the positive correlations between chestnut and eucalyptus honey with hydrocarbons, esters, ketones, proline, alanine, serine, and aspartic acids and the negative correlations between the terpenes, asparagine, histidine, and lysine are shown. The Sulla honeys showed an opposite relationship. Citrus showed a positive correlation between alcohols, aldehydes, furans, valine, isoleucine, leucine, phenylalanine, and tirosine; Acacia honey showed a negative correlation with the same molecules of citrus. Figure 3 shows that two amino acids (threonine and glutamic acid) cannot be related to product characteristics since they are located in the center of the graphical display.

The model was a correlation expressed as a Pearson index between 17 amino acids and 130 volatile compounds grouped in different chemical families (aldehydes, furans, hydrocarbons, ketones, alcohols, and esters). The data obtained showed strong positive linear correlations between aldehydes and isoleucine, valine, leucine, and phenylalanine. Furans are correlated with isoleucine, leucine, and phenylalanine; hydrocarbons with serine, glutamic acid, and aspartic acid; and ketones with serine, alanine, glutamine, histidine, asparagine, and lysine. Alcohols were more associated with tyrosine than esters with arginine. Proline, tryptophan, and threonine showed poor correlations with all the classes of aroma compounds. This research opens avenues for enhanced traceability and identification of honey based on its amino acid and aromatic composition. This feasibility study shows the excellent potentiality of the model as a modern analytical tool for a rapid multicomponent analysis of honey quality indicators. The utility of the developed predictive model is closely tied to the feasibility of obtaining information about honey using much simpler analytical protocols. For example, as demonstrated, gaining insights into the amino acid composition of honey requires the use of rather complex analytical methods due to the nature of the molecules and especially the matrix. Conversely, the analysis of the aromatic profile is straightforward: it does not demand excessive sample preparation and is also relatively cost-effective and environmentally friendly, as it does not require solvents. Through correlation modeling, the analysis of the aromatic profile has the potential to not only define the sensory quality of the product but also the amino acid composition. Additionally, it can provide information about the botanical origin, given that many amino acids are considered markers for identifying specific types of honey.

### Future Recommendations

The outcomes achieved underscore the potential of utilizing the combined analysis of amino acid composition and aromatic profile, in conjunction with statistical analysis, as a valuable tool for identification purposes and for establishing a means of robust tracking and traceability. To our knowledge, this is the first time that a similar investigation has been carried out on the honeys. Since the properties of honey are strictly dependent on various factors, conducting a larger number of analyses by examining samples of different honeys, both in terms of botanical and geographical origin, will be necessary to establish the stability of the analytical method. Future research will be further extended to other Italian, European, and non-European honeys with the aim of confirming the validity of the developed method and also applying the same method to assess its feasibility in recognizing geographical origins.

## 4. Conclusions

In conclusion, this study delved into the multifaceted world of honey. In this study, which represents a feasibility study, a predictive model was built. A chemometric approach was employed to decipher the intricate relationships between aroma variability data and amino acids linked to botanical origin. To build the model, an analysis of 25 honeys of different botanical origins collected in the Calabria region (South Italy) was performed. Through hydrophilic interaction chromatography (HILIC) coupled with mass spectrometry, the amino acid profile was determined. By headspace solid phase micro-extraction (HS-SPME) with gas chromatography mass spectrometry (GC-MS), the aromatic profile was extracted. A robust statical approach was used to determine the Person’s correlation between amino acid profiles and aromatic compounds in various types of honey. Finally, various types of honeys, their amino acid profiles, and their aroma characteristics were correlated by PLS analysis. The results revealed a strong linear association between specific amino acids and various classes of aromatic compounds, shedding light on the potential impact of amino acids on honey’s aromatic profile. Further investigations can extend this methodology to explore additional honey varieties, thus contributing to a comprehensive understanding of the intricate interplay between amino acids and aroma compounds in this prized natural product.

## Figures and Tables

**Figure 1 foods-12-03284-f001:**
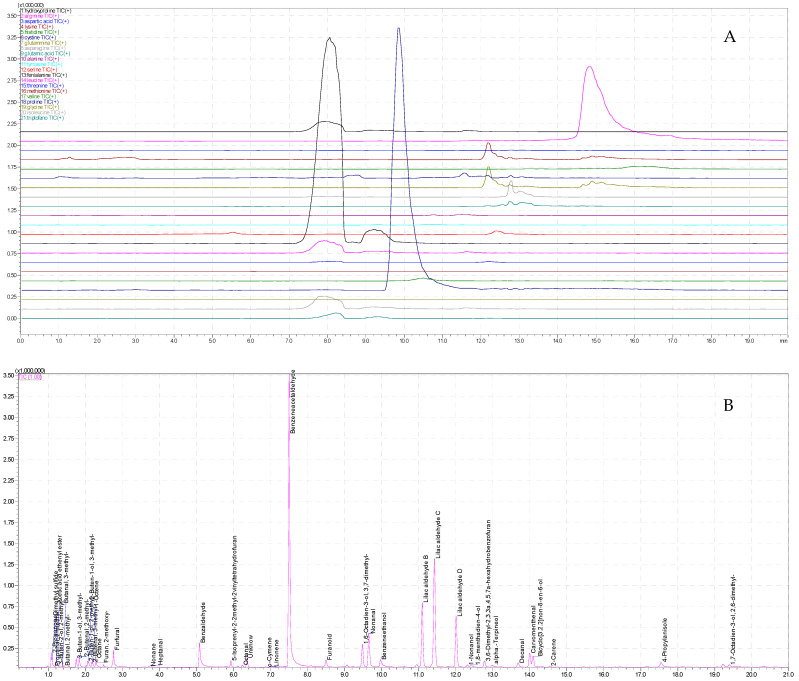
Citrus honey: (**A**) Amino acids profile by HILIC-UPLC-ESI-MS/MS (from the top down: Hydroxyproline, Arginine, Aspartic Acid, Lysine, Histidine, Cysteine, Glutamine, Asparagine, Glutamic Acid, Alanine, Tyrosine, Serine, Phenylalanine, Leucine, Threonine, Methionine, Valine, Proline, Glycine, Isoleucine, Tryptophan) (**B**) Volatile profile by HS-SPME-GC-MS of Citrus Honey.

**Figure 2 foods-12-03284-f002:**
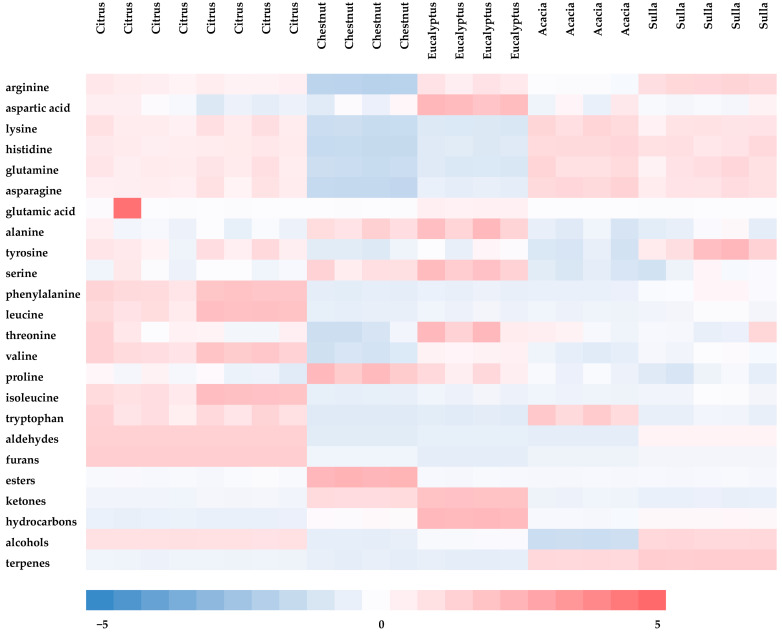
Heat map of honey samples showing area of compounds by amino acids and volatile classes (blue color means low quantity and red color is high quantity, relatively).

**Figure 3 foods-12-03284-f003:**
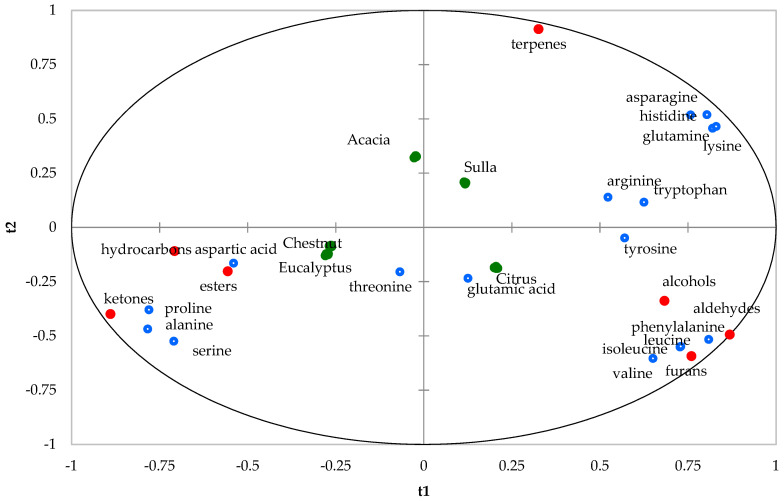
Correlation circle of the honey type (green). aromatic classes (red) and amino acids (blue) (t_1_; t_2_).

**Table 1 foods-12-03284-t001:** MRM amino acid transitions.

Compound	Precursor (*m*/*z*)	Product (*m*/*z*)	Dwell (msec)	Q1 (V)	Ce	Q3 (V)
Glycine	76.1	30	13	−12	−11	−12
	76.1	47.9	13	−12	−10	−18
Tyrosine	182	136.1	13	−11	−15	−28
		165.3			−12	−20
Aspartic Acid	134.2	74.1	13	−14	−16	−30
		87.9	13	−14	−13	−19
Lysine	146.7	130.2	100	−30	−15	−15
		84.1	100	−30	−27	−30
Seryne	105.9	60.1	13	−10	−12	−24
	106	42	13	−21	−22	−17
	106.1	83	100	−10	−19	−21
Proline	116.1	70.1	30	−19	−16	−30
Valine	118.1	72.1	8	−22	−12	−15
		55.1			−22	−23
	158.9	118	100	−16	−9	−13
Methionine	150.2	56.1	8	−10	−17	−12
	150.2	61			−24	−24
Fenilalanine	165.6	120.1	8	−30	−14	−26
		103.1	8	−30	−27	−22
Arginine	175.1	70.1	13	−11	−22	−29
		60.1	13	−11	−14	−25
		116	13	−11	−15	−24
Leucine	131.9	86.1	8	−27	−12	−22
		41.6			−30	−18
Isoleucine	131.9	86.1	8	−13	−12	−19
		69			−17	−27
Threonine	120.2	74	13	−12	−12	−15
		102.2		−24	−13	−22
Glutamic Acid	147.8	83.9	22	−14	−15	−17
		130		−29	−14	−15
Alanine	90.1	44.1	13	−17	−13	−21
	90.1	44.9			−31	−18
Histidine	155.6	110.1	13	−30	−14	−23
		93			−23	−18
		93.1			−23	−11
Asparagine	133	74	22	−14	−15	−30
	133.1	87.1		−12	−12	−18
Glutammine	147.1	84	22	−16	−15	−22
		130			−24	−26
Tryptophan	204.9	118.1	8	−22	−24	−25
	204.9	118.1	8	−22	−10	−21
Cistyne	241.2	74.1	13	−10	−26	−29
		152			−12	−28
	240.8	122	100	−11	−16	−26
Hydroxyproline	132.2	74.2	100	−17	−23	−29
	123.1	86.1		−15	−14	−18

**Table 2 foods-12-03284-t002:** Pearson correlations among amino acids and aroma compounds (classes) in honey samples.

Aroma Compounds Classes	Amino Acids	Pearson Correlation Coefficient	*p* Value	Correlation Curve
Aldehydes	Isoleucine	0.931	<0.0001	y = 0.827x − 0.0151
	Valine	0.922	<0.0001	y = 0.810x − 0.0151
	Leucine	0.932	<0.0001	y = 0.8276x − 0.0151
	Phenylalanine	0.972	<0.0001	y = 0.8631x − 0.0151
Furans	Isoleucine	0.982	<0.0001	y = 0.8188x − 0.0852
	Leucine	0.982	<0.0001	y = 0.8186x − 0.0852
	Phenylalanine	0.983	<0.0001	y = 0.8194x − 0.0852
Hydrocarbons	Serine	0.793	<0.0001	y = 0.7894x + 0.0461
	Glutamic Acid	0.941	<0.0001	y = 0.9552x − 0.1389
	Aspartic Acid	0.888	<0.0001	y = 0.8971x − 0.1019
Ketones	Serine	0.905	<0.0001	y = 0.9127x + 0.0116
	Alanine	0.929	<0.0001	y = 0.99364x + 0.0116
	Glutamine	−0.964	<0.0001	y = −0.9715x + 0.0116
	Histidine	−0.979	<0.0001	y = −0.99867x + 0.0116
	Lysine	−0.958	<0.0001	y = −0.9655x + 0.0116
	Asparagine	−0.964	<0.0001	y = −0.9714x + 0.0116
Alcohols	Tyrosine	0.927	<0.0001	y = 1.1056x + 0.0364
Esters	Arginine	−0.98	<0.0001	y = −1.00631x + 0.0992

**Table 3 foods-12-03284-t003:** Distribution of amino acids concentration for Calabria unifloral honey (mg/kg).

	*Citrus* (*n* = 8) *		Chestnut (*n* = 4)		Eucalyptus (*n* = 4)		Acacia (*n* = 4)		Sulla (*n* = 5)	
Amino Acids	MV	SD	MV	SD	MV	SD	MV	SD	MV	SD
Alanine (Ala)	171.06	15.92	240.13	14.52	273.41	25.80	146.93	14.87	162.49	18.59
Serine (Ser)	84.83	5.09	99.92	5.10	112.03	5.24	74.66	2.53	81.70	7.02
Proline (Pro)	1101.80	159.11	2062.37	160.65	1563.91	190.93	1050.58	82.48	922.90	135.79
Valine (Val)	684.54	59.76	304.88	21.75	510.53	12.53	373.41	24.24	441.27	22.26
Threonine (Thr)	46.06	5.65	32.89	4.71	59.77	8.74	43.17	4.12	42.31	7.72
Leucine (Leu)	988.19	277.33	59.46	10.57	142.74	33.32	137.22	16.39	223.54	60.38
Isoleucine (Ile)	1005.88	279.90	52.94	8.29	140.93	34.92	142.43	15.77	221.58	62.94
Asparagine (Asn)	826.29	72.74	197.73	12.44	463.69	21.31	1038.45	36.56	912.67	60.04
Aspartic acid (Asp)	212.73	33.52	208.32	29.51	373.77	13.59	226.69	37.83	219.02	15.68
Lysine (Lys)	315.08	35.21	24.11	4.92	87.95	6.52	387.67	23.46	323.08	35.17
Glutamine (Gln)	201.27	16.93	16.36	3.32	56.44	5.48	240.67	18.07	221.65	34.91
Glutamic acid (Glu)	174.64	270.06	73.17	1.89	143.30	6.98	76.85	1.70	79.16	5.40
Histidine (His)	169.71	5.53	34.69	1.69	76.50	3.97	209.21	3.82	192.92	12.20
Phenylalalnine (Phe)	1866.58	359.49	120.68	23.66	223.13	48.51	229.06	8.37	666.61	152.22
Arginin (Arg)	161.67	10.23	15.28	2.31	180.88	13.69	125.01	3.81	214.88	9.10
Tyrosine (Tyr)	380.47	47.37	263.61	17.57	317.55	33.08	240.29	21.83	472.73	71.48
Tryptophan (Trp)	36.03	6.22	1.51	0.01	2.88	0.79	44.25	6.28	8.22	2.60

* number of samples; MV: mean value; SD: standard deviation.

## Data Availability

The data used to support the findings of this study can be made available by the corresponding author upon request.

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
