# Peer review of "A Predictive Model to Correlate Amino Acids and Aromatic Compounds in Calabrian Honeys"

_foods, 2023, doi:10.3390/foods12173284_

Round 1
Reviewer 1 Report
Overall, it's a very informative research work with good interpretation of results. This work should attract the readers' focus. However, the introduction section is unnecessarily long. Please abridge this section. Provie a section namely " Previous researches and bridge between recent trends and the past" . Include a section on "Future recommendations" after discussion section. Please include quantitative analysis of amino acids in the manuscript. Without the quantitative analysis, the novelty is much hampered. There are some typos. Check the grammatical integrity as well.
Overall, it's a very informative research work with good interpretation of results. This work should attract the readers' focus. However, the introduction section is unnecessarily long. Please abridge this section. Provie a section namely " Previous researches and bridge between recent trends and the past" . Include a section on "Future recommendations" after discussion section. Please include quantitative analysis of amino acids in the manuscript. Without the quantitative analysis, the novelty is much hampered. There are some typos. Check the grammatical integrity as well.
Author Response
Dear Reviewer,
Thank you for your comments and suggestions on our manuscript entitled: “A predictive model to correlate amino acids and aromatic compounds in Calabrian honeys”.
We are grateful for the opportunity to answer the questions related to our study submitted to Foods. The manuscript was modified according with the reviewer and editor suggestions. The corrections or specific answers are listed below point by point.
Some sections of the text have been completely modified following a request made by the editor to reduce the repetition rate of the manuscript.
- The introduction section is unnecessarily long. Please abridge this section
AUTHORS: the whole introduction section has been rewritten and abridged
- Provie a section namely " Previous researches and bridge between recent trends and the past".
AUTHORS: this section was added at the end of introduction
- Include a section on "Future recommendations" after discussion section.
AUTHORS: it has been done
- Please include quantitative analysis of amino acids in the manuscript. Without the quantitative analysis, the novelty is much hampered.
AUTHORS: The table with amino acids quantitative analysis was added, and the text includes comments related to the quantitative analysis
- There are some typos. Check the grammatical integrity as well
AUTHORS: The grammatical integrity has been checked.
Reviewer 2 Report
Manuscript Number: foods-2552483
Manuscript Title: A predictive model to correlate amino acids and aromatic compounds in Calabrian honeys
In this work, a complete analysis of 25 honeys, five types of unifloral honey collected in Calabria region (South Italy) was performed on the correlation between volatile compound and amino acids. The study is interesting but there are some problems to be solved. My comments and suggestions are listed below.
Comments:
1.What are the innovations and significance of this study? The authors found that some volatile components showed a good linear relationship with amino acid content, while others showed a poor linear relationship. In addition, volatile components and amino acids can be affected by many factors, such as the source of honey, processing methods and so on. So, do the authors' results apply to all situations?
2. Can the author's experimental results be used to identify the varieties and trace the origin? If the author can provide some relevant data, the article may be more innovative.
3.Line 145: H2O→H2O.
4.Line 161: ml→mL.
5.Conclusion: the text of future research perspectives is suggested to be slightly supplemented.
6. Some references are irregular in format and need to be proofread.
Author Response
Dear Reviewer,
Thank you for your comments and suggestions on our manuscript entitled: “A predictive model to correlate amino acids and aromatic compounds in Calabrian honeys”.
We are grateful for the opportunity to answer the questions related to our study submitted to Foods. The manuscript was modified according with the reviewer and editor suggestions. The corrections or specific answers are listed below point by point.
Some sections of the text have been completely modified following a request made by the editor to reduce the repetition rate of the manuscript.
- What are the innovations and significance of this study? The authors found that some volatile components showed a good linear relationship with amino acid content, while others showed a poor linear relationship. In addition, volatile components and amino acids can be affected by many factors, such as the source of honey, processing methods and so on. So, do the authors' results apply to all situations? Can the author's experimental results be used to identify the varieties and trace the origin? If the author can provide some relevant data, the article may be more innovative.
AUTHORS: this is a preliminary study to develop the analytical and statical methods with the aim to build a predictive model for correlating amino acids profile to the aromatic compounds using chemometric technique. The results unveiled strong linear association between specific amino acids and various classes of aromatic compounds.
Since the properties of honey are strictly dependent on various factors, conducting a larger number of analyses by examining samples of different honeys, both in terms of botanical and geographical origin, will be necessary to establish the stability of the analytical method. Future research will be further extended to other Italian, European and not honeys with the aim of confirm the validity of the developed method, also with the aim to apply the same method to assess its feasibility in recognizing geographical origins.
- Line 145: H2O→H2
AUTHORS: Done and the whole manuscript has been checked
- Line 161: ml→mL.
AUTHORS: Done and the whole manuscript has been checked
- Conclusion: the text of future research perspectives is suggested to be slightly supplemented.
AUTHORS: the future research perspectives were rewritten
- Some references are irregular in format and need to be proofread
AUTHORS: the references were checked and modified.
Reviewer 3 Report
Change the sign (†) of corresponding author to (*)
Does not define at the beginning of the manuscript what PLS means
Homogenize throughout the manuscript 20°C apart, since in some 20°C are together.
Change the letter “l” in liters to the capital letter “L” in mL, review throughout the manuscript
In some paragraphs use the symbol "–" and in others "-" to separate numerical values, homogenize with the first
Table 1. The names of the amino acids change to lowercase (first letter is capitalized and the rest are lowercase” and the same with the column titles. In the same table, homogenize the values with one or two after the decimal point.
Table 2. The names of the amino acids and classes of aromatic compounds, change to lowercase (first letter capital and the rest lowercase" and the same with the column titles.
In figure 1. The name of some amino acids is cut off, correct it, and change the first letter to uppercase
The space between lines in "Conclusions" is not the same, please homogenize
In the references, update it according to the citation form of the journal, it is also observed that it is not the same space between line and line, please homogenize.
Line 105, 116: Hydrophilic with lowercase letter “H”
Line 109, 214: Change GCMS to GC-MS
Line 145, 160: The number “2” of the water molecule is with a subscript
Line 162: Separate 5min, like 5 min. Revise the entire manuscript with other values and units found together as on line 163 (0.22 um) and change “um” to the correct symbol for micro, as found on line 165.
Line 198: L/min is separated, leave “L/min” together
Line 206: Is “(GC-MS_QP2010) the underscore correct?
Line 224: what is the version of the program?
Author Response
Dear Reviewer,
Thank you for your comments and suggestions on our manuscript entitled: “A predictive model to correlate amino acids and aromatic compounds in Calabrian honeys”.
We are grateful for the opportunity to answer the questions related to our study submitted to Foods. The manuscript was modified according with the reviewer and editor suggestions. The corrections or specific answers are listed below point by point.
Some sections of the text have been completely modified following a request made by the editor to reduce the repetition rate of the manuscript.
- Change the sign (†) of corresponding author to (*)
AUTHORS: It has been done
- Does not define at the beginning of the manuscript what PLS means
AUTHORS: PLS was defined
- Homogenize throughout the manuscript 20°C apart, since in some 20°C are together.
AUTHORS: Done and the whole manuscript has been checked
- Change the letter “l” in liters to the capital letter “L” in mL, review throughout the manuscript Done.
AUTHORS: Done and the whole manuscript has been checked
- In some paragraphs use the symbol "–" and in others "-" to separate numerical values, homogenize with the first
AUTHORS: Done
- Table 1. The names of the amino acids change to lowercase (first letter is capitalized and the rest are lowercase” and the same with the column titles. In the same table, homogenize the values with one or two after the decimal point.
Table 2. The names of the amino acids and classes of aromatic compounds, change to lowercase (first letter capital and the rest lowercase" and the same with the column titles.
In figure 1. The name of some amino acids is cut off, correct it, and change the first letter to uppercase
AUTHORS: it has been done
- The space between lines in "Conclusions" is not the same, please homogenize
AUTHORS: I’m sorry! The space now was homogenized
- In the references, update it according to the citation form of the journal, it is also observed that it is not the same space between line and line, please homogenize.
AUTHORS: The references was checked according with the form of journal. Number 32 and 33 have the same space between lines but i don’t know because they appear diffrent from the others.
- Line 105, 116: Hydrophilic with lowercase letter “H”
AUTHORS: it has been done
- Line 109, 214: Change GCMS to GC-MS
AUTHORS: it has been done
- Line 145, 160: The number “2” of the water molecule is with a subscript
AUTHORS: Done and the whole manuscript has been checked
- Line 162: Separate 5min, like 5 min. Revise the entire manuscript with other values and units found together as on line 163 (0.22 um) and change “um” to the correct symbol for micro, as found on line 165.
AUTHORS: Done and the whole manuscript has been checked
- Line 198: L/min is separated, leave “L/min” together
AUTHORS: Done and the whole manuscript has been checked
- Line 206: Is “(GC-MS_QP2010) the underscore correct?
AUTHORS: It’s correct because it’s the name of the instrument
Line 224: what is the version of the program?
AUTHORS: XLSTAT Vs. 2022.4.5. It has been added in the text.
Round 2
Reviewer 2 Report
The author has made a good revision of this paper, and it is recommended to accept.
Author Response
we are grateful for your comment